# Atypical Plasma Cell Leukemia Mistaken for Acute Leukemia: A Case Report

**DOI:** 10.3390/medicina60081351

**Published:** 2024-08-20

**Authors:** Irena Seili-Bekafigo, Emina Torlakovic, Tajana Grenko Malnar, Marija Stanić Damić, Željko Prka, Koviljka Matušan Ilijaš, Ita Hadžisejdić

**Affiliations:** 1Department of Cytology, Faculty of Medicine, University of Rijeka, Clinical Hospital Center Rijeka, 51000 Rijeka, Croatia; 2Department of Pathology and Laboratory Medicine, College of Medicine, University of Saskatchewan, Saskatoon, SK S7N 5A2, Canada; emina.torlakovic@saskhealthauthority.ca; 3Department of Hematology, Faculty of Medicine, University of Rijeka, Clinical Hospital Center Rijeka, Krešimirova 42, 51000 Rijeka, Croatia; 4Department of Hematology, University Hospital Dubrava, 10000 Zagreb, Croatia; zeljkoprka@yahoo.com; 5Department of Pathology, Faculty of Medicine, University of Rijeka, Clinical Hospital Center Rijeka, 51000 Rijeka, Croatia; koviljka.matusan@medri.uniri.hr (K.M.I.);

**Keywords:** plasma cell leukemia, acute leukemia, immunophenotype, immunohistochemistry, flow cytometry

## Abstract

The patient we present here had many clinical, morphological, and laboratory findings characteristic of acute leukemia. During the course of the disease, the diagnosis changed from acute leukemia to chronic small B-cell lymphoproliferative disease, a blastoid variant of mantle cell lymphoma, and finally to atypical plasma cell leukemia. Atypical plasma cell leukemia is a rare condition with aggressive biological behavior. Our patient relapsed a short time after achieving complete remission, in spite of aggressive therapy and autologous stem cell transplantation. During relapse, it was possible to morphologically identify malignant cells as being of plasma cell origin, although immature and atypical. Atypical plasma cell leukemia presents a diagnostic challenge as it may mimic other neoplasms both morphologically and clinically. It is also recognized that plasma cell neoplasm immunophenotype may not be entirely specific for its lineage where common diagnostic biomarkers are applied by immunohistochemistry or flow cytometry. Where this is the case, only focused investigation for plasma cell lineage will be more informative. This patient has unusual clinical presentation, a nondescript morphology of the circulating plasma cells, as well as an immunophenotype, detected by the initial panels used for flow cytometry and immunohistochemistry, that was not entirely specific for plasma cells. Such cases present a good reminder of the diagnostic complexity of atypical plasma cell leukemia and emphasize that plasma cell differentiation needs to be interrogated in cases where the initial work-up shows unusual results.

## 1. Introduction

Plasma cell leukemia (PCL) is a rare and extremely aggressive variant of plasma cell dyscrasia with a dismal prognosis [1]. PCL can arise de novo (primary PCL) or as a leukemic transformation in relapsed/refractory multiple myeloma (MM) (secondary PCL). The incidence of newly diagnosed PCL is estimated at 0.04 cases per 100,000 persons/year [2], while approximately 2.0–4.0% of patients diagnosed with MM progress into this entity [3]. The clinical presentation may encompass features observed in MM, as well as those seen in other leukemias, including leukocytosis, anemia, thrombocytopenia, infections, and hepatosplenomegaly. A frequently observed complication is extramedullary infiltration of solid organs and the central nervous system [1]. In 2021, the diagnostic criteria were revised from the presence of 20% to 5% or more of circulating clonal plasma cells (PC) in peripheral blood smears [4]. The morphology of plasma cells in this entity displays a different level of maturation, with the majority of cases exhibiting lymphoplasmacytoid or plasmablastic characteristics. When presenting with immature and poorly differentiated plasma cells, there is a need for extended immunophenotyping to confirm the diagnosis. Two commonly used markers, CD38 and CD138, show comparable expression in MM and PCL. In contrast to MM, PCL demonstrates a less differentiated phenotype with lower levels of CD9, CD56, CD71, CD117, and HLA-DR, but higher levels of CD20, CD23, CD28, CD44, and CD45 [5,6,7]. Molecular cytogenetics performed by FISH show a predominance of t(11;14) (47%), del(17p) (28%), and t(14;16) (12%) in PCL [8].

In this report, we discuss the case of a 52-year-old female patient who was diagnosed with atypical plasma cell leukemia after it was previously thought to be either an acute lymphoblastic leukemia, chronic small B-cell lymphoproliferative disease, or a blastoid type of mantle cell lymphoma.

## 2. Case Report

A 52-year-old Caucasian woman without any past medical history was admitted to hospital because of spontaneous bruising and significantly increased leukocytes. Physical examination revealed ecchymoses and hematomas of the lower extremities without lymphadenopathy, hepatomegaly, or splenomegaly. Laboratory findings showed normocytic normochromic anemia, thrombocytopenia, leukocytosis with lymphocytosis, a high level of atypical cells, and immunoparesis, while coagulation tests were within normal ranges (Table 1).

Examination of the peripheral blood smear demonstrated 74% of blast-like cells (Figure 1). Bone marrow cytology identified 69% of atypical cells with irregular, often cleaved nuclei and dense chromatin that were mostly negative for myeloperoxidase (<1% positive cells), non-specific esterase (<2% positive cells), and periodic acid-Schiff (<1% positive cells) on cytochemistry (Figure 2). Since there was a strong suspicion of acute leukemia, the patient was transferred to a specialized center for further evaluation and treatment of this type of hematological disease. Using immunophenotyping, up to 56% of aberrant CD45-CD38+ cells that lacked typical myeloid and lymphoid markers (CD19 1%; kappa+CD19+ 0.5%; lambda+CD19+ 0.5%; CD56 8%), including those of plasma cells (CD138 0%), were discovered in the bone marrow. Immunophenotyping analysis of blood was not performed.

Surprisingly, repeated bone marrow biopsy suggested low-grade small B-cell lymphoproliferative disease with the aberrant phenotype and a relatively low Ki-67 proliferation index of about 20%. Acute leukemia was excluded based on bone marrow biopsy immunophenotyping. No chromosomal aberrations were detected with conventional karyotyping.

Additional work-up discovered an abnormal serum-free light-chain ratio due to high concentrations of lambda light chain (sFLC-κ = 3.89 mg/L, sFLC-λ = 4060 mg/L, κ/λ = 0.0009), while serum electrophoresis showed a monoclonal band in the gamma region. Monoclonal gammopathy was confirmed by serum immunofixation, which tested positive for the lambda chain. Whole-body MSCT showed bone remodeling of the whole skeleton and minimal bilateral pleural effusions. Although renal function was preserved, a 24 h urine collection revealed nephrotic syndrome (total daily protein = 3770 mg) with a significant proportion of lambda light chains (uFLC-κ = 5.5 mg/L, uFLC-λ = 2080 mg/L, κ/λ = 0.00264) and slight albuminuria (440 mg). Due to high troponin (T = 30, NTproBNP = 3488 ng/L), peripheral microvoltage on the ECG, along with PR prolongation and slow progression of the R-wave in precordial leads the patient was also evaluated by a cardiologist who excluded acute coronary syndrome and confirmed preserved systolic and diastolic heart function with minimal pericardial effusion. The magnetic resonance of the heart was performed and revealed changes suggestive of AL amyloidosis including mild myocardial thickening of the anterolateral wall and subendocardial imbibition in the circumferential pattern. A rectal mucosal biopsy was performed, but no amyloid accumulation was found. Considering the new findings, a bone marrow biopsy was performed once again, only this time pathohistological analysis suggested a blastoid variant of mantle cell lymphoma (negative: CD3, CD5, CD20, CD79a, PAX5, CD10, Bcl6, IgM, TdT, CD34, CD117, CD138, SOX11, kappa; positive: Cyclin D1, MUM1, CD38, lambda; Ki-67 = 50%). Considering the inconclusive diagnosis obtained by the previous diagnostic process, the sample was sent for external hematopathology consult. Considering that clinical presentation and laboratory findings were mostly suggestive of plasma cell dyscrasia and that the patient’s condition required the initiation of treatment, our patient was started on pre-phase steroid treatment (methylprednisolone 1 mg/kg) and then continued with an alternating VCD/PAD regimen (Table 2). In the meantime, the results of the bone marrow aspirate and biopsy from the external consultation arrived, indicating atypical plasma cell leukemia with strong expression of cyclin D1, which usually suggests t(11;14). Due to technical reasons, FISH analysis of bone marrow cells was not performed. The tumorous cells in this case were negative for CD27 and CD138, which is consistent with an immature phenotype and a more aggressive clinical course. Additional Congo red staining for amyloid was negative. Following four treatment cycles, hematological reevaluation showed a stringent complete remission, with absent aberrant clone in control immunophenotyping of the bone marrow (CD19+ 6%; CD27+ 49%; CD56 14%; CD38 36%; CD81 88%; CD117 7%; CD138 0%), accompanied by the full resolution of proteinuria. Given the patient’s good overall condition and preserved cardiac ejection fraction, she remained a candidate for autologous stem cell transplantation. After the mobilization with high-dose cyclophosphamide, stem cell harvesting was successful, so the patient underwent tandem autologous stem cell transplants (conditioning high-dose melphalan 200 mg/m^2^). The whole procedure was without complications and the patient was in remission. We initiated maintenance therapy with bortezomib and dexamethasone every two weeks. Unfortunately, only two months later, our patient developed persistent pain in the upper left abdominal quadrant and noticed a right inguinal lymph node enlargement (5 cm). Blood tests identified mild macrocytic anemia (Hb 115 g/L, MCV 98.5 fL), thrombocytopenia (315 × 10^9^/L), hypercalcemia (2.75 mmol/L), positive serum immunofixation, and a sudden increase in lambda light-chain levels (sFLC-κ = 3.3 mg/L, sFLC-λ = 2600 mg/L), which indicated a relapse of the disease. CT scans detected multiple large tumor masses in the abdomen. An ultrasound-guided biopsy of the pancreatic mass showed atypical plasma cells with lambda restriction (Figure 3). Bone marrow biopsy was also consistent with the disease relapse, revealing 30% of abnormal plasma cells (Ki-67 = 95%) (Figure 4).

Several different regimens were applied in a short period of time, the Dd-PACE regimen and KRd regimen with the addition of bendamustine (Table 2), but without success—the disease was resistant to treatment and progressive with the appearance of new extramedullary foci of the disease. During that time, the patient developed progressive hemorrhagic pleural effusion and severe infective complications. Our patient passed away a year and a half after the onset of symptoms.

## 3. Discussion

The identification of acute leukemia (AL), based on morphology of the peripheral blood (PB) smear and bone marrow (BM) aspirate, is usually straight forward. Often, it is possible to further differentiate between AL with lymphoblastic (ALL) or myeloid (AML) differentiation by using histochemistry. After the cytomorphological and cytochemical analysis of the BM aspirate, further classification of ALs is established by immunophenotyping and molecular genetics [9,10,11,12]. In the fifth edition of the WHO classification for myeloid and histiocytic/dendritic tumors, morphology alone as a diagnostic premise to make a diagnosis of AL is removed [13]. This patient presented with history typical for AL, which paired well with the finding of 74% of atypical, blast-like cells in PB and 69% in BM. Since all cytochemical stainings used for the classification of ALs were negative, the findings were still broadly consistent with ALL or even undifferentiated AML. The immunophenotype of the atypical cells, assessed by flow cytometry from the same BM specimen, was CD45−CD38+CD138−, lacking myeloid or lymphoid markers. In the other hospital, PB and BM cytology and BM biopsy were repeated, and the findings were different than the initial ones from our hospital. Atypical cells looked morphologically more mature in the second PB smear, BM aspirate, and biopsy. They were CD25+, only rarely CD20, CD79a, CD10, and PAX5+, lambda+/kappa−, CD2, CD3, CD99, Tdt, CD117, CD34, CD31, MPO, CD56, CD138 negative, and Ki-67 was only 20%. The diagnosis of low-grade small B-cell lymphoproliferative disease with aberrant phenotype was made. Repeated BM biopsy performed a month later, again at our hospital, showed diffuse marrow infiltration with atypical cells. The immunoprofile was consistent with blastoid mantle cell lymphoma with leukemic spread. The Ki-67 increased to 50%. Due to discrepant findings in different institutions, the case was referred to an external expert, where the diagnosis of atypical plasma cell leukemia was made. This plasma cell leukemia was negative for CD27 and CD138, which is a rare finding and associated with aggressive biological behavior. That was proven true because our patient relapsed after a short time, despite aggressive therapy and autologous stem cell transplantation. During relapse it was possible to morphologically identify malignant cells as being of plasma cell origin, although immature and atypical. The Ki-67 was 95% at that time, which explains such an adverse clinical course.

This patient was given several different diagnoses during the relatively short duration of her disease. It started with the diagnosis of AL, the second diagnosis was small B-cell lymphoproliferative disease, then the blastic variant of mantle cell lymphoma, and finally atypical plasma cell leukemia. The morphology of plasma cells in PCL can vary from mature, readily recognizable forms, to atypical, immature blastoid cells, and may represent a diagnostic challenge. Plasma cell leukemia with blastoid morphology is a very rare entity and is characterized by aggressive clinical behavior, as was the case with our patient. Fang and Xu described a similar case, in which the patient presented with leukocytosis and blasts in PB, thus mimicking acute leukemia. Furthermore, Toriyama et al. described a case of oligosecretory plasma cell leukemia with atypical morphology of leukemic cells, which is also very difficult to differentiate from other leukemic diseases [14,15,16]. During the wide spread of the disease, atypical immature plasma cells can be found in unexpected; for example, in the urine sediment [17].

## 4. Conclusions

Atypical plasma cell leukemia presents a diagnostic challenge as it may mimic other neoplasms both morphologically and clinically. Plasma cell neoplasm immunophenotype may not be entirely specific for its lineage where common diagnostic biomarkers are applied by immunohistochemistry or flow cytometry. The complexity of malignant hematological diseases gives rise to the need for applying a broad panel of diagnostic biomarkers for immunophenotyping at the moment of diagnosis, especially where the results are unusual. Finally, the need for close collaboration between laboratory and clinical specialists (hematologists) should be pointed out, especially when dealing with atypical disease presentation.

## Figures and Tables

**Figure 1 medicina-60-01351-f001:**
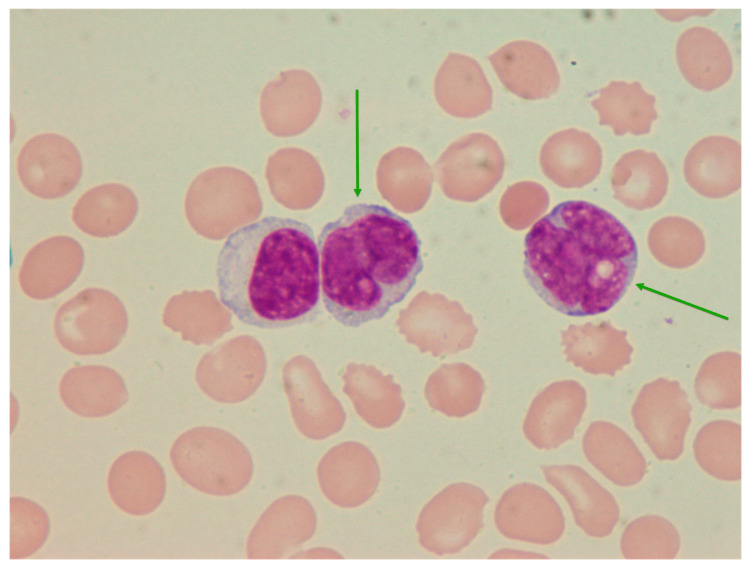
Peripheral blood smear at the moment of diagnosis. May–Gruenwald–Giemsa, ×1000. Two atypical cells (green arrows) with irregular nuclei and prominent nucleoli. There is no definite morphologic evidence of plasma cell differentiation.

**Figure 2 medicina-60-01351-f002:**
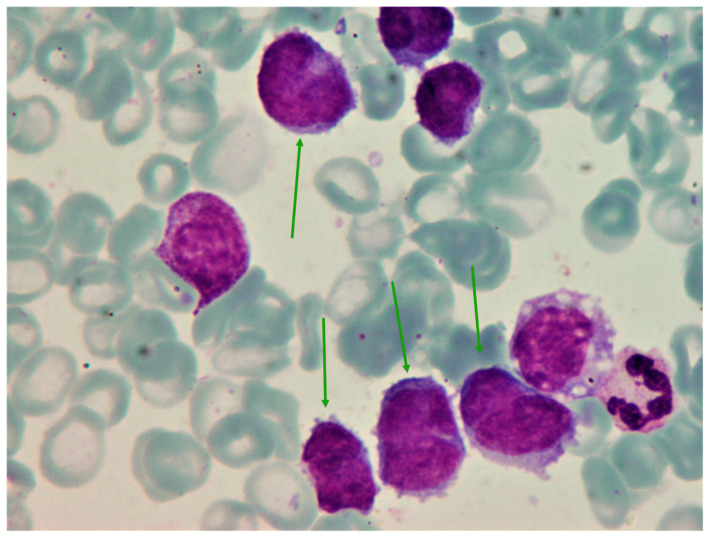
Bone marrow aspirate at the moment of diagnosis. May–Gruenwald–Giemsa, ×1000. Several atypical cells (green arrows) with blast-like morphology.

**Figure 3 medicina-60-01351-f003:**
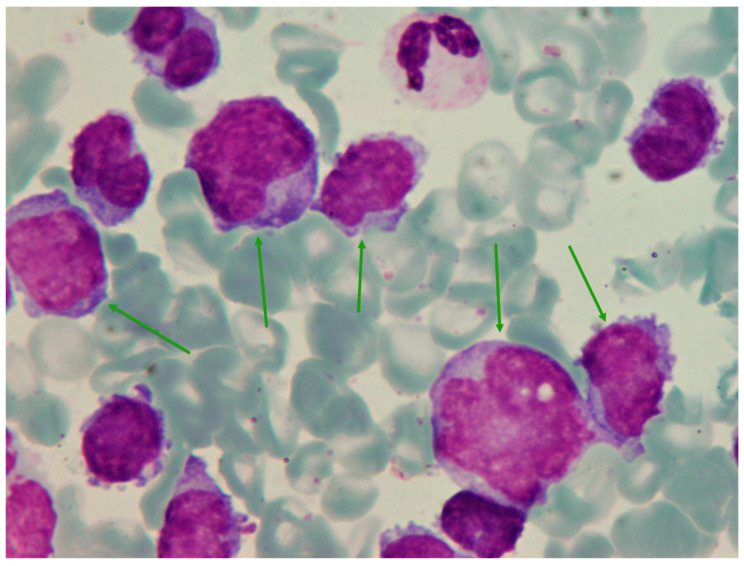
Fine-needle aspirate of the large abdominal mass at relapse. May–Gruenwald–Giemsa, ×1000. Many atypical cells (green arrows) are present.

**Figure 4 medicina-60-01351-f004:**
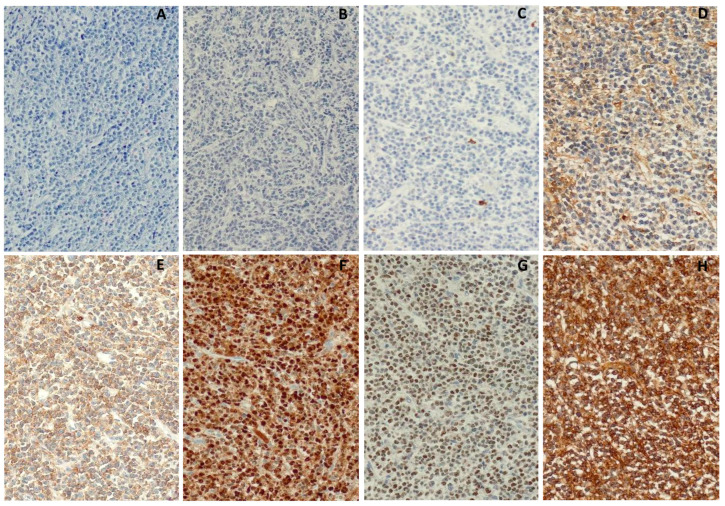
Bone marrow biopsy shows diffuse infiltration composed of small- to medium-sized cells with high nuclear/cytoplasmic ratio ((**A**), Giemsa stain, ×200). Immunohistochemistry for PAX5 ((**B**), ×200), CD138 ((**C**), ×200), and kappa light chains ((**D**), ×200) is negative while positive for CD38 ((**E**), ×200), cyclin D1 ((**F**), ×200), MUM1 ((**G**), ×200), and lambda light chains ((**H**), ×200).

**Table 1 medicina-60-01351-t001:** The laboratory findings at hospital admission.

	Patient’s Values	Units	Reference Values
Erythrocytes	3.04	×10^12^/L	3.86–5.08
Hemoglobin	93	g/L	119–157
MCV	92.3	fL	83.0–97.2
Thrombocytes	51	×10^9^/L	158–424
Leukocytes	21.90	×10^9^/L	3.4–9.7
Neutrophils	2.15	×10^9^/L	2.06–6.49
Lymphocytes	6.53	×10^9^/L	1.19–3.35
Atypical cells	12.35	×10^9^/L	0
Ig A	<0.2	g/L	0.7–4.0
Ig M	<0.2	g/L	0.4–2.3
Ig G	1.2	g/L	7.0–16.0
PT	1.23	1	0.70–1.40
APTT	33.87	s	25.00–40.00
Fibrinogen	3.88	g/L	1.80–4.00

**Table 2 medicina-60-01351-t002:** Applied treatment regimens.

**1st LINE alternate: VCD (21-day cycle) ×2**	**PAD (21-day cycle) ×2**
bortezomib 1.3 mg/m^2^ SC D1, D4, D8, D11 Cyclophosphamide 500 mg IV D1, D8, D15 Dexamethasone 20 mg IV D1, D2, D4, D5, D8, D9, D11, D12	doksorubicin IV 10 mg/m^2^ D1–D4 bortezomib 1.3 mg/m^2^ SC D1, D4, D8, D11 deksametazon 40 mg IV, D1–D4
**2nd LINE: Dd-PACE (28-day cycle) ×2**	
daratumumab 16 mg/kg IV D1, D8, D15, D22 cisplatin 10 mg/m^2^ IV D1–4 doxorubicin 10 mg/m^2^ IV D1–4 cyclophosphamide 400 mg/m^2^ IV D1–4 etoposide 40 mg/m^2^ IV D1–4 Dexamethasone 20 mg IV D1, D2, D8, D9, D15, D16, D22, D23	
**3rd LINE: KRd + bendamustine 28-day cycle ×1**	
karfilzomib 20 mg/m^2^ IV D1–2, 27 mg/m^2^ D8–9, D15–16 lenalidomide 25 mg PO D1–21 dexamethasone 40 mg IV D1, D8, D15 bendamustine 90 mg/m^2^ IV D1–2	

## Data Availability

Reported diagnostic tests can be found at the Department of Cytology and Department of Pathology of the Clinical Hospital Center, Rijeka, in the archive materials. Clinical data can be found in the Department of Hematology of the Clinical Hospital Center Rijeka, Croatia.

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
