# Peer review of "Atypical Plasma Cell Leukemia Mistaken for Acute Leukemia: A Case Report"

_medicina, 2024, doi:10.3390/medicina60081351_

Round 1

Reviewer 1 Report

Comments and Suggestions for Authors

This case report presents an unusual etiology, challenging differential diagnosis and puzzling clinical features of a 52 year old caucasian woman with PCL. However the report can be strengthened by including the below dataset:

1) Inclusion and exclusion criteria adopted for such patients

2) Flow cytometry data of the immunophenotyping analysis of blood or bone marrow.

3) Tabular column of the treatment regimes followed

4) Possible causes for the differences in BM biopsy analysis of other hospitals from the initial diagnosis. 

Reviewer 2 Report

Comments and Suggestions for Authors

I have reviewed the case report titled "Atypical plasma cell leukemia for acute leukemia; a case report” with great interest. This report effectively highlights the diagnostic complexity of atypical plasma cell leukemia and emphasizes the importance of collaboration between laboratory and clinical specialists for accurate diagnosis. The introduction is adequate, the case report is well-written, and the discussion is consistent with the study.

However, there are a few issues that need to be addressed and modified by the authors prior to publication. I have noted the following comments:

 - The description of Figure 2 on page 3, lines 84-85 should be placed on the next page to avoid confusion with one photo and two descriptions.

 - Similarly, on page 5, lines 140-144, the same issue regarding the placement of descriptions needs to be addressed.

Round 2

Reviewer 1 Report

Comments and Suggestions for Authors

The answers to the raised points are satisfactory. This manuscript could be accepted for publication.